

# Optimizing sowing time and weather conditions for enhanced growth and seed yield of chia (*Salvia hispanica* L.) in semi-arid regions

CB Harisha[1,*], KM Boraiah[1,*], PS Basavaraj[1], Hanamant M. Halli[1], Ram Narayan Singh[1], Jagadish Rane[1,2], K Sammi Reddy[1], GR Halagundegowda[3], Amresh Chaudhary[1,4], Arvind Kumar Verma[5], Y Ravi[5], Honnappa Asangi[6] and E Senthamil[7]

[1] ICAR-National Institute of Abiotic Stress Management, Baramati, Maharashtra, India
[2] ICAR-Central Institute of Arid Horticulture, Bikaner, Rajasthan, India
[3] Central Silk Board, Bangalore, Bangalore, Karnataka, India
[4] ICAR-Central Soil Salinity Research Institute, Karnal, Haryana, India
[5] ICAR-National Research Centre on Seed Spices, Ajmer, Rajasthan, India
[6] ICAR-Indian Institute of Spice Research Regional Station, Appangala, Karnataka, India
[7] University of Agricultural Sciences, Dharwad, Karnataka, India
[*] These authors contributed equally to this work.

Corresponding authors
CB Harisha, Harisha.B@icar.gov.in
Hanamant M. Halli, hmhalli4700@gmail.com

## ABSTRACT

**Background**. Climate influenced weather events, especially during the flowering, grain filling, and maturity stages, can adversely influence crop yield and quality. Therefore, understanding how the phenological behaviour and yield potential of new crops such as chia are influenced by weather and sowing dates is crucial for maximizing crop yield. This study aimed to assess the impact of sowing dates on the flowering behaviour, and yield attributes of chia morphotypes, as well as to identify optimal weather conditions for achieving higher yields.

**Methods**. The study was conducted during 2021–22 and 2022–23 and consisted of fifteen sowing windows from 1st July to 1st February (at 15 days intervals), with two chia morphotypes (white and black seed) arranged in a replicated split-plot design. Phenological events, flowering characters and seed yield traits were recorded regularly. Weather parameters at the experimental location (Maharashtra, India) were recorded.

**Results**. The results revealed that weather conditions such as relative humidity (RH) and rainfall favoured the flowering phenology, yield attributes, and seed yield of chia, whereas maximum temperature ($T_{max}$), bright sunshine hours, and accumulated growing degree days had negative effects. Weather parameters significantly influenced the chia seed yield during the cropping period: RH (positive, $R^2 = 86.1\%$), $T_{max}$ (negative, $R^2 = 67.4\%$), rainfall (positive, $R^2 = 52.9\%$), and diurnal temperature range (negative, $R^2 = 74.9\%$). Black-seeded chia morphotypes consistently produced higher seed yields (10.8% greater) and better yield-contributing traits compared to white types across various sowing dates. The maximum chia seed yield (811–793.1 kg ha$^{-1}$) was achieved with sowing dates between August 1st and September 1st in this semi-arid region of India. The performance of chia was good under congenial weather

conditions, including relative humidity (~67–72%), maximum temperature (~30–31 °C), day length (<12.0 h), rainfall (~200–350 mm), and accumulated growing degree days (~1,521–1,891). The present study findings can help identify the best suitable regions for chia cultivation by revealing relationships between the performance of chia morphotypes and weather conditions.

## INTRODUCTION

Climate change-induced weather events adversely influence the yield and quality of oilseeds by altering crop-growing conditions at both regional and national levels (*Attia et al., 2021*). The global average yield of major oilseed crops such as sunflower, soybean, and canola has plateaued over the last several years (*Attia et al., 2021*; *Ray et al., 2019*). In the last few decades, the import of oilseeds has increased tremendously in the Indian subcontinent due to decreased productivity of major oilseed crops (*Brassicaceae*) (*Jingar et al., 2023*). An average healthy adult intakes about 20–35% of their calories through oil and fats. The human body is unable to synthesize two essential fatty acids: alpha-linolenic and linoleic acids (*Saini & Keum, 2018*). Thus, this causes ever-increasing pressure on global food and nutritional security and determines the Sustainable Development Goals (SDG2, zero hunger) (*Halli et al., 2024*). Therefore, these two essential fatty acids must be directly obtained from healthy sources like fish, and oilseed crops such as chia to reduce the risk of cardiovascular diseases and high blood pressure (*Kris-Etherton & Krauss, 2020*).

In this context, chia (*Salvia hispanica* L.) is an important crop belonging to the *Lamiaceae* family with high nutritional and medicinal values, thriving well in tropical and subtropical climates (*Capitani et al., 2013*). Besides, chia oil can also be used for industrial purposes such as a stabilizer and binder in food processing (*Felisberto et al., 2015*; *Pathak et al., 2015*), and as an anti-corrosive agent. Apart from its higher protein content, chia seeds contain a notable amount of fixed oil (20.3% to 38.6%), prominently featuring $\alpha$–linolenic acid (55%) and linoleic acid (19%) (*Attia et al., 2023*; *Ayerza & Coates, 2011*). The well-balanced profile of essential amino acids makes chia a preferred ingredient for the development of health-oriented products; hence, it is often referred to as a "superfood" (*Fernandes et al., 2020*). Accordingly, the consumer tendency to choose food crops like chia, nutri-millets, and grain amaranth is increasing due to multiple health benefits and to combat malnutrition. Consequently, in India, chia cultivation extends across many central and southern states to meet the increasing demand for balanced edible oil and industrial applications. In 2023, the global market for chia was valued at US$ 203 million, and further market insights anticipate a cumulative growth rate of at least 7%, reaching US$ 390 million by 2033 (*ChiaSeedMarket, 2024*). Because of its suitability under resource-scarce conditions (water, poor soils, and nutrients) of tropical and subtropical regions, the area under chia cultivation is gradually

increasing in many states of the country (*Harisha et al., 2023*). However, limited technical information is available on cultivation practices, such as optimum sowing time, weather relation with flowering behaviour and yield traits in semi-arid regions (*Attia et al., 2023*; *Jingar et al., 2023*).

In recent years, deviated weather events such as rainfall, temperature, and relative humidity have altered crop performances, prompting farmers to adopt sowing windows that may not be optimum for crop performance in general. Similarly, in the case of chia, varied sowing windows from July–August to mid-winter December–January result in dwindling responses in terms of flowering, maturity, seed yield, and oil quality (*Karim, Ashrafuzzaman & Hossain, 2015*; *Ram et al., 2024*). Chia seed yield is highly responsive to sowing dates, yielding 150 kg ha$^{-1}$ in December sowing and 354 kg ha$^{-1}$ in October sowing under Indian conditions (*Guttedar et al., 2023*). These variations could be predominantly credited to the wide range of prevailing weather conditions (temperature, relative humidity, and rainfall), especially in photosensitive crops (*Ayerza & Coates, 2009*; *Hirich, Choukr-Allah & Jacobsen, 2014*). Flower induction in chia requires temperatures between 20-−30 °C, annual rainfall between 500–1,000 mm, and a photoperiod of less than 12 h (*Jamboonsri et al., 2012*). Suboptimal photoperiods can lead to reduced reproductive phases and increased vegetative growth (*Baginsky et al., 2016*). For example, early sowing in June or July encounters high temperatures and long day lengths initially, extending the growth period or accumulating higher heat units, which leads to enhanced vegetative biomass but decreased seed yield and oil content in chia (*Brandan, Izquierdo & Acreche, 2022*; *Benetoli da Silva et al., 2020*). A positive relation was observed between pre-flowering duration and verticillaster flower weight. In contrast, longer duration leads to more flower dry weight and seed yield in chia. However, the study is limited to growing degree days and photoperiod, and the effect of weather parameters before and after flowering was not considered to explain the yield related traits *Brandan, Curti & Acreche (2020)* and similarly, delayed sown chia experiences initial cooler temperatures and shorter days, followed by hot and dry conditions, which lead to premature floral initiation and shorten the vegetative phase. Therefore, timely sowing is a basic requirement to provide ideal weather conditions for determining the growth and yield of chia (*Baginsky et al., 2016*). A favourable day length and weather conditions during the flowering and seed setting stages of chia can optimize the yield and oil quality (*Lobo et al., 2011*).

Apart from climate, diverse morphotypes of chia respond differently to environmental conditions and sowing times (*Benetoli da Silva et al., 2020*). Both white and black-seeded chia types differ in their growth, yield, and oil content (33.8% and 32.7%, respectively) as reported by *Suri, Passi & Goyat (2016)*. However, many studies did not explore how chia morphotypes respond to varying sowing dates, photoperiods, temperatures, and relative humidity concerning growth, pre and post-flowering behaviour, and seed development in semi-arid conditions. The growth dynamics and distribution of assimilates in plants strongly depend on temperature, relative humidity, and moisture availability (*Silva et al., 2017*). Limited previous studies have investigated the performance of either white or black-seeded chia morphotypes under limited sowing dates and overlooked remaining sowing windows. Yet, no studies have deciphered the impact of wider sowing windows

(fifteen sowing dates at intervals of 15 days) in a year on flowering phenology and maturity in two chia morphotypes. The lack of knowledge on how chia morphotypes behave in terms of phenology and seed yield in response to prevailing weather parameters limits the ability to maximize seed yield. Therefore, choosing the ideal sowing time to achieve better synchronized flowering and high seed yield is a primary requirement for any grower or plant breeder. Understanding crop phenology and its relationship with weather helps plant breeders advance in generation and enables growers to assess yield potential. Such information on how sowing dates and weather parameters influence chia seed and oil yield is crucial for characterizing photoperiod sensitivity and guiding the selection of new niches for chia intensification, thereby reducing climate-induced weather uncertainties to meet the increasing market demand for quality vegetable oil and addressing SDG 13 (climate action). Thus, we hypothesize that sowing dates favouring weather conditions influence flowering phenology and yield attributes of chia types, and the interaction of temperature, relative humidity, and rainfall would optimize vegetative and reproductive phases. Therefore, a two-year field study was planned to determine the effect of varied sowing windows and weather conditions on flower phenology, maturity, and seed yield of chia morphotypes and also to understand the association between key weather parameters in determining the chia seed yield.

## MATERIALS AND METHODS

### Weather details of the study location

Field trials were conducted for two consecutive years (2021–22 and 2022–23) at ICAR–National Institute of Abiotic Stress Management (NIASM), Baramati, Pune, Maharashtra, India (Fig. S1). The study site is positioned at 18.15850556°N and 74.50085556°E at an elevation of 570 m above sea level (MSL). This region falls within the hot and semi-arid zone of the Deccan Plateau region, which is known as the water scarcity zone. The mean maximum and minimum temperature of the region was 31.2 °C and 21.9 °C, respectively. The region receives an average annual precipitation of 576 mm, a major portion (75%) is received between August and October (Harisha et al., 2023). The annual open-pan evaporation rate of the region is 1,965 mm, which is three times more than annual rainfall. The detailed weather parameters for the cropping seasons (2021–22 and 2022–23) are outlined in Fig. 1 and Table S1. The weather data on maximum temperature (Tmax), minimum temperature (Tmin), bright sunshine hours (BSS), open pan evaporation, rainfall and relative humidity (RH) for the location during the cropping season was obtained from the weather observatory of ICAR-NIASM, Baramati.

### Soil details of the experimental site

The soil type of the experimental site was shallow basaltic with 81.9% sand, 10.4% silt, and 7.5% clay exhibiting low water holding capacity (Rajagopal et al., 2018). The chemical properties of the soil were pH (7.48), an electrical conductivity (0.21 dS m$^{-1}$), a moderate level of organic carbon (6.5 g kg$^{-1}$), low available nitrogen (81.2 kg ha$^{-1}$), phosphorus (3.6 kg ha$^{-1}$ as $P_2O_5$), and potassium (80.0 kg ha$^{-1}$ as $K_2O$).

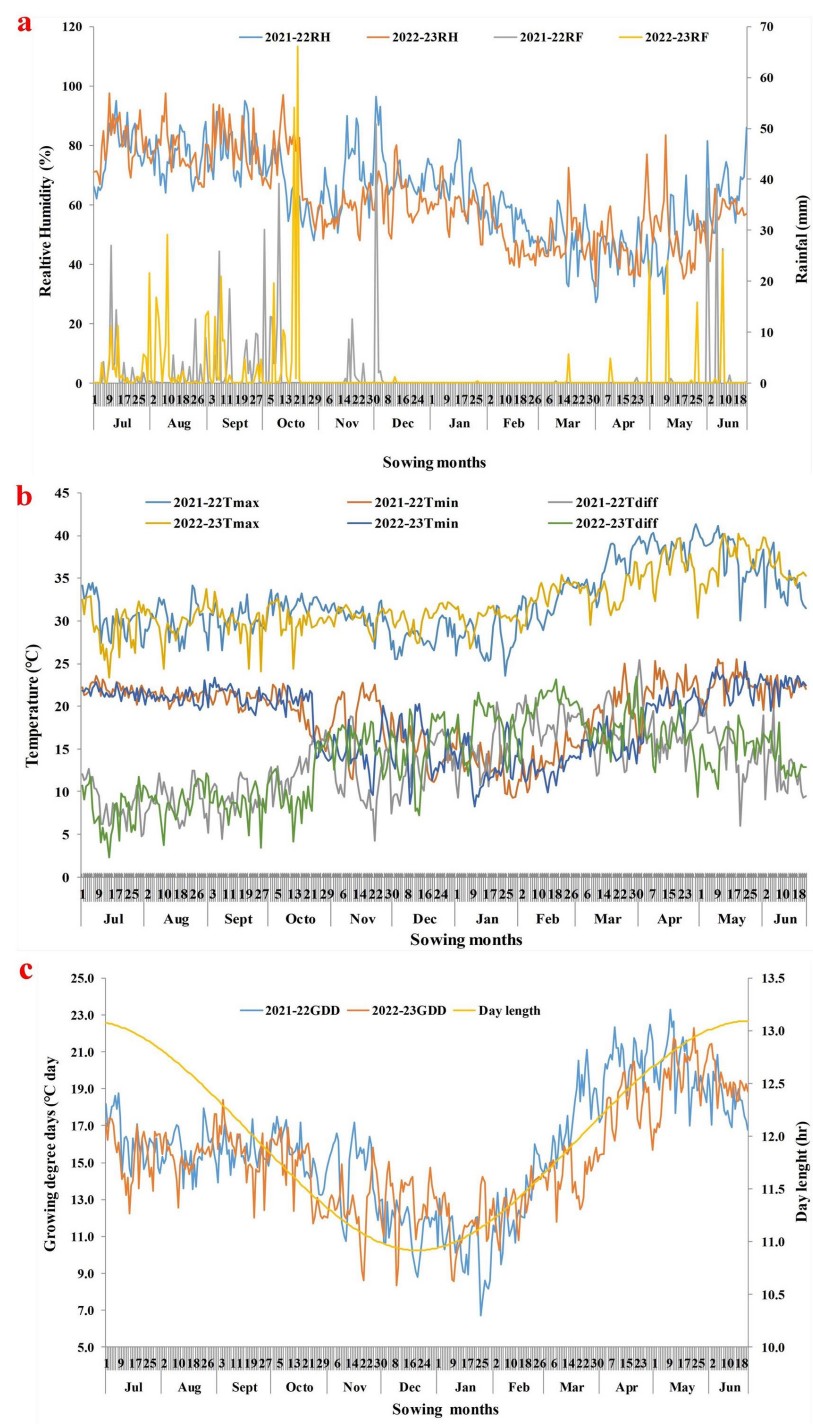

**Figure 1  Prevailing weather parameters during chia cropping period (sowing to maturity) in both years.** (A) Relative humidity and rainfall during the cropping period; (B) mean maximum and mean minimum temperature during cropping period; (C) growing degree days and day length during cropping period.

## Experimental details and crop management

The experiment consists of two factors; dates of sowing and chia morphotypes were laid out in split-plot design with three replications. Fifteen dates of sowing (S1; 1st July, S2; 15th July, S3; 1st August, S4; 15th August, S5; 1st September, S6; 15th September, S7; 1st October, S8; 15th October, S9; 1st November, S10; 15th November, S11; 1st December, S12; 15th December, S13; 1st January, S14; 15th January, and S15; 1st February) were treated as main factor and two chia types (white and black) as subfactor. The plots of size 3 m × 2.5 m were prepared for sowing the seeds of chia types (2.5 kg ha$^{-1}$) after mixing with sand in 60 cm wider rows. Subsequently, excess and weak plants were removed by retaining one healthy and maintained a uniform distance of 20 cm between plants within rows. Recommended nutrients (N:P$_2$O$_5$:K$_2$O at 90:60:75 kg ha$^{-1}$) were applied through fertilizers such as urea, di-ammonium phosphate, and muriate of potash (*Harisha et al., 2023*). The full dose of P$_2$O$_5$ and K$_2$O, and 50% of N was applied during field preparation as a basal, whereas the remaining 50% of N (45 kg/ha) was top dressed through urea in three splits at 30 days after sowing (DAS) (15 kg/ha), 45 DAS (15 kg/ha), and 60 DAS (15 kg/ha). The urea was applied in band placement five cm away from plants and light raking was done to mix it with soil. The chia was cultivated under rainfed conditions with supplemental irrigation. During extended dry spells, supplemental irrigation was scheduled based on soil drying and visible crop moisture deficit symptoms during both the rainy and winter seasons, at a depth of five cm at weekly intervals using a drip system. Weeds were controlled by manual hand weeding. However, the crop remained unaffected by pests and diseases during both cropping periods.

## Measurement of chia morphological parameters and phenology

Chia growth attributes such as plant height and dry biomass production were recorded at harvest from five randomly selected plants, separately in each treatment. Floral characters such as days to flower bud appearance (FBA), completion of flowering, and maturity were recorded from randomly selected five plants as per the procedure outlined by *Brandan, Curti & Acrechea (2019)*. A day to 50% flowering was recorded treatment wise when 50% of plants opened their first flower. Likewise, growing degree days (GDD), also called heat unit, accumulated up to maturity were calculated for each sowing date as suggested by *Nuttonson (1957)*.

$$GDD = \frac{T_{Max} + T_{Min}}{2} - T_{base}.$$

T$_{max}$ is maximum temperature, T$_{min}$ is minimum temperature, T$_{base}$ is base temperature (10 °C) *Ayerza & Coates (2009)*.

Likewise, heat use efficiency (HUE) indicates the capacity of a plant to produce yield per unit of heat used. HUE of the chia crop was calculated using the formula suggested by *Singh & Khushu (2012)*.

$$HUE\left(kg\ ha^{-1}\ {}^{\circ}C^{-1}\ day^{-1}\right) = \frac{Grain\ yield\ (kg\ ha^{-1})}{Accumulated\ GDD\ ({}^{\circ}C\ day)}$$

### Seed yield and yield attributes of chia

Yield determinants of chia such as the number of spikes per plant, spike length, seed yield per spike and 1,000 seed weight were recorded from five randomly selected plants from each treatment *(Harisha et al., 2024)*. Then, seed yield was determined by recording the seed weight from fifty plants in the plot of 7.5 m$^2$ and sun dried for 3–4 days to attain moisture content of $7 \pm 0.5\%$ and expressed in kg ha$^{-1}$. Likewise, plot wise dry biomass yield was determined from randomly selected five plants after sun drying for 2–3 days followed by oven drying at 63 °C for 72 h to attain constant weight and expressed as dry biomass kg ha$^{-1}$. Later, the harvest index (HI) was calculated based on the seed and biological yield of chia.

$$\text{HI (\%)} = \frac{\text{Grain yield (kg ha}^{-1})}{\text{Dry Biomass (kg ha}^{-1})}$$

Later, grain filling duration (GFD) was calculated considering the number of days between 50% flowering and physiological maturity. Similarly, the grain filling rate was calculated by dividing seed yield with grain filling duration as explained by *Sattar et al. (2023)* in wheat.

### Statistics

Before conducting an analysis of variance, the data recorded on various growth, phenology, and yield parameters of chia during both years was tested for normality by the Shapiro–Wilk test using the PROC UNIVARIATE procedure in SAS 9.3 (SAS Institute, Inc., Cary, NC, USA). Then, normal data was subjected to analysis of variance (ANOVA) using the mixed model (proc GLIMMMIX in SAS v 9.3). Chia morphotypes, year, and sowing dates were considered as fixed effects and replications as random effects. *Post-hoc* test was conducted to compare the difference ($\alpha = 0.05$) using Tukey's honest significant difference (HSD) test. Further, Pearson's correlation coefficient was used to describe the association between weather parameters (T$_{max}$, T$_{min}$, RH, accumulated GDD, bright sunshine hours, and rainfall), *vs* grain yield, days to flower bud appearance, flowering duration, and maturity *(Gomez & Gomez, 1984)*. To interpret multi-environment (sowing dates × weather parameters × chia types) interaction, principal component analysis (PCA) biplot analysis was carried out using R software (version 4.2.3) *(Gopinath et al., 2021)*.

## RESULTS

### Chia growth and floral phenology

Growth determinants such as plant height and biomass accumulation in chia morphotypes differed significantly ($p < 0.05$) across sowing dates (Figs. 2A–2B). Among chia types, black seeded plants were found to be more vigorous with greater height (119.6 cm) and biomass accumulation (2,883.9 kg ha$^{-1}$) compared with white seeded plants (117.3 cm, and 2,662.2 kg ha$^{-1}$ respectively). Regarding fifteen sowing dates, early sowing (S1: 1st July, S2: 15th July, S3: 1st August) demonstrated the highest plant height (199.1 cm, 195.1 cm, and 185.3 cm respectively,), and biomass production (4,294.2–4,021.9 kg ha$^{-1}$) compared to other sowing dates (Figs. 2A–2B). Whereas delayed sowing after S3 up to S15

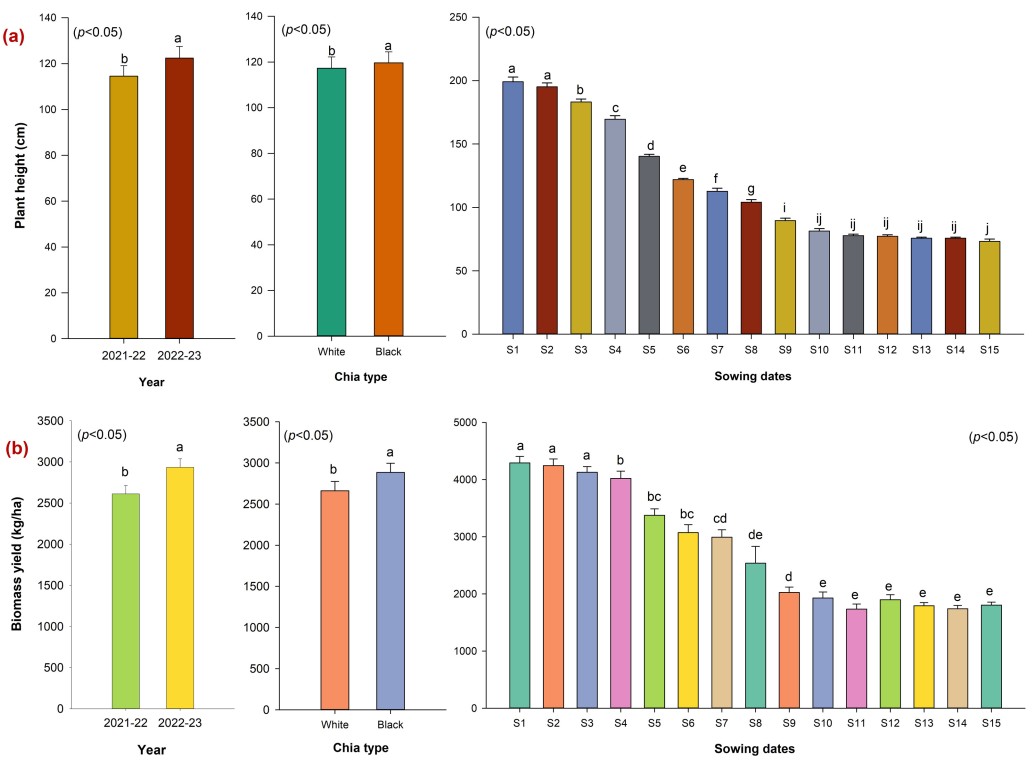

**Figure 2 Plant height and biomass accumulation in chia morphotypes as influenced by sowing dates (2021–22 and 2022–23).** W, White seed chia; B, Black Seed chia; S1, 1st July; S2, 15th July; S3, 1st August; S4, 15th August; S5, 1st September; S6, 15th September; S7, 1st October; S8, 15th October; S9, 1st November; S10, 15th November; S11, 1st December; S12, 15th December; S13, 1st January; S14, 15th January; S15, 1st February.

conspicuously reduced the plant height and biomass accumulation (1,735.1–1,899.4 kg ha$^{-1}$) in chia types.

Similarly, floral phenological events such as days to flower bud appearance (FBA), days to 50% flowering, days to completion of flowering, days to maturity and flowering duration have responded ($p < 0.05$) to dates of sowing (Table 1 and Table S2). Notably, flowering phenology did not differ among white and black seeded chia morphotypes. Whereas, early sown plants (S1 and S2) took more days to FBA (70.5–78.2), and it was drastically reduced to 35.0 days in late sown conditions (S7: 1st October). Further delay in sowing after S8: 15th October to S15: 1st February gradually delayed the FBA (54.8 days). Similarly, days to 50% flowering, days to complete flowering and days to maturity followed a similar trend as that of FBA (Table 1). The flowering duration was significantly delayed in late sown conditions (S13 to S15; 63.6 to 77.5 days) over other sowing dates. The shortest flowering duration of 47 days was observed in S8 and S9 sowing conditions. Moreover, early sown conditions (S1 to S4) enhanced the grain filling duration (39.8 to 41.8 days) with a decreasing trend up to S11 and a subsequent increase up to S15. Across years of cultivation, the second year (2022–23) noticed maximum plant height (122.4 cm), and biomass accumulation

**Table 1  Influence of sowing windows on flower phenology and maturity of chia morphotypes.**

| Treatments | Days to flower bud appearance | Days to 50% plants flowering | Days to completion of flowering | Days to maturity | Flowering duration (days) | Grain filling duration (days) |
|---|---|---|---|---|---|---|
| Year (Y) | | | | | | |
| 2021–22 | 49.6[a]† | 78.8[a] | 102.6[b] | 114.1[b] | 53.0[b] | 35.2[b] |
| 2022–23 | 48.8[b] | 78.8[a] | 106.2[a] | 115.3[a] | 57.4[a] | 36.5[a] |
| *P* value | <0.001 | NS | <0.001 | <0.001 | <0.001 | 0.015 |
| Date of sowing (DOS) | | | | | | |
| S1–1st July | 78.2[a] | 103.3[a] | 134.8[a] | 143.1[a] | 56.5[d] | 39.8[ab] |
| S2–15th July | 70.5[b] | 94.2[b] | 127.3[b] | 135.6[b] | 56.6[d] | 41.4[a] |
| S3–1st August | 62.7[c] | 85.1[c] | 117.4[c] | 125.9[d] | 54.6[de] | 40.8[ab] |
| S4–15th August | 54.8[d] | 72.6[e] | 106.3[e] | 114.4[e] | 51.4[e−g] | 41.8[a] |
| S5–1st September | 45.2[g] | 70.8[e] | 98.3[f] | 105.9[f] | 51.3[e−g] | 35.1[de] |
| S6–15th September | 39.4[i] | 62.0[f] | 89.9[g] | 98.8[g] | 50.5[f−h] | 36.7[cd] |
| S7–1st October | 35.0[j] | 62.7[f] | 83.8[h] | 95.0[h] | 48.7[g−i] | 32.3[ef] |
| S8–15th October | 35.2[j] | 62.5[f] | 81.3[i] | 93.7[h] | 46.0[i] | 31.1[f] |
| S9–1st November | 35.2[j] | 64.0[f] | 83.0[h] | 94.3[h] | 47.0[hi] | 30.3[f] |
| S10–15th November | 38.9[i] | 64.5[f] | 86.0[h] | 95.3[h] | 47.0[hi] | 30.8[f] |
| S11–1st December | 41.0[h] | 78.5[d] | 94.9[f] | 105.3[f] | 43.9[d−f] | 26.7[g] |
| S12–15th December | 44.8[g] | 79.7[d] | 97.4[f] | 114.6[e] | 52.5[ef] | 34.9[de] |
| S13–1st January | 48.6[f] | 86.7[c] | 112.3[d] | 125.0[d] | 63.6[c] | 38.3[bc] |
| S14-15th January | 52.5[e] | 94.2[b] | 121.3[c] | 132.1[c] | 68.8[b] | 37.9[b−d] |
| S15–1st February | 54.8[d] | 102.1[a] | 132.3[a] | 142.3[a] | 77.5[a] | 40.1[ab] |
| *P* value | <0.001 | <0.001 | <0.001 | <0.001 | <0.001 | <0.001 |
| Chia type (S) | | | | | | |
| White | 49.0[a] | 78.9[a] | 104.1[a] | 114.6[a] | 55.1[a] | 35.8[a] |
| Black | 49.3[a] | 78.7[a] | 104.6[a] | 114.9[a] | 55.3[a] | 36.0[a] |
| *P* value | NS | NS | NS | NS | NS | NS |
| Interaction effect | | | | | | |
| Y × S | NS | NS | NS | NS | NS | NS |
| Y × DOS | <0.001 | <0.001 | <0.001 | <0.001 | 0.930 | <0.001 |
| S × DOS | NS | NS | NS | NS | NS | NS |
| Y× S × DOS | NS | NS | NS | NS | NS | NS |

**Notes.**
†Means followed by the same letter (s) within the column are not significantly differed ($p < 0.05$).
The different letter denoted in superscript in each column shows the significant difference between the treatments.

(2,934.7 kg ha$^{-1}$) with delayed flowering duration, grain filling duration and maturity (115.3 days).

## Relation between prevailing weather parameters and flowering phenology of chia

Weather conditions during the vegetative phase (germination to bud appearance) strongly influenced the flowering phenology of chia (Fig. 3A). The Pearson's correlation suggested that FBA exhibited a positive correlation with day length ($r = 0.7$), accumulated GDD
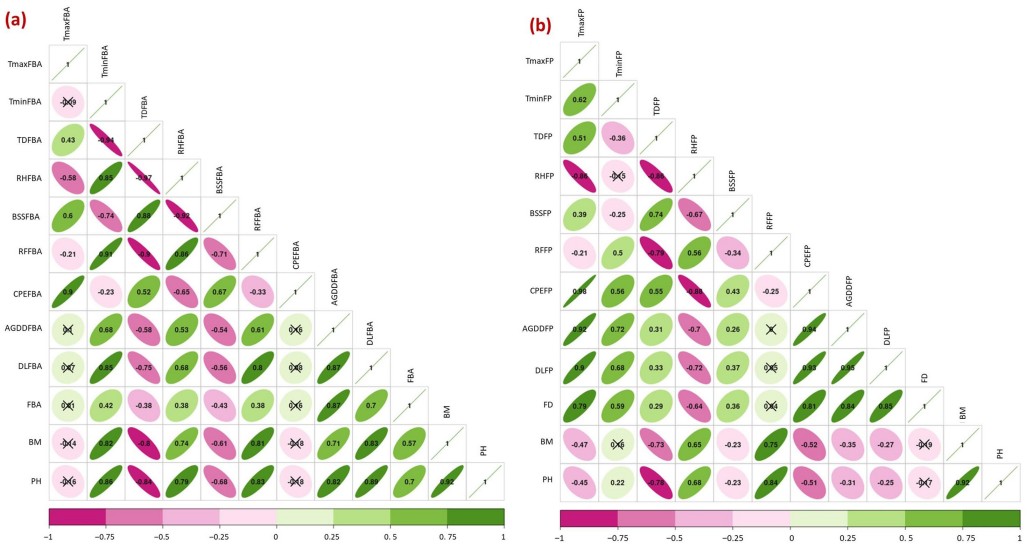

**Figure 3** **Relation between weather parameters and flowering phenology of chia (A) during flower bud appearance (FBA), (B) during flowering phase (FP).** Tmin, minimum temperature; Tmax, maximum temperature; TD, diurnal temperature difference; RH, relative humidity; BSS, bright sun shine hours; RF, total rainfall; CPEFBA, cumulative pan evaporation; AGDD, accumulated growing degree days; DL, day length; FBA, days to flower bud appearance; PH, plant height; BM, biomass; FD, flowering duration; FP, flowering phase. *Cells marked with x are non-significant at $p = 0.01$.

($r = 0.87$), $T_{min}$ ($r = 0.42$), and RH ($r = 0.38$). While FBA was negatively related to diurnal temperature difference ($T_{diff}$) ($r = -0.38$) and bright sunshine hours (BSS) ($r = -0.43$). The flowering duration had a positive correlation with day length ($r = 0.85$), $T_{max}$ ($r = 0.79$), $T_{min}$ ($r = 0.59$), and accumulated GDD ($r = 0.84$) prevailed during flowering phase (flower initiation to completion). However, flowering duration was negatively correlated with RH prevailing during the flowering phase ($r = -0.64$) (Fig. 3B).

## Yield attributes and seed yield of chia

Yield attributes of chia morphotypes responded to sowing dates during two years of investigation (Table 2). Black seeded chia types produced more spikes per plant (30.3), longer spike length (17.99 cm), higher 1,000 seeds weight (1.15 g), HUE (0.37 kg ha$^{-1}$ °C$^{-1}$ day$^{-1}$), grain filling rate (17.8 kg ha$^{-1}$ day$^{-1}$), and seed yield (564.6 kg ha$^{-1}$) compared to white types. While white seeded morphotypes maintained a greater harvest index (21.32%) across sowing dates. Within sowing dates, treatments (S3–S6; 1st August–15th September) maintained a greater number of spikes, spike length (20.1–21.71 cm), 1,000 seeds weight (1.15–1.16 g), and seed yield (741.0–811.0 kg ha$^{-1}$) with greater HUE, grain filling rate, and HI. In contrast, delayed sowing after S7 to S15 adversely influenced the HUE, grain filling rate, and seed yield of chia morphotypes. Regarding year effect, the first year (2021–22) recorded a superior number of spikes, 1,000 seed weight, HUE, grain filling rate, and seed yield (579.2 kg ha$^{-1}$) over 2022–23 (Table 2). Further, the triple interaction between sowing dates, chia types, and year showed a significant effect on seed yield and HUE. In
**Table 2  Yield attributes and heat use efficiency (HUE) of chia morphotypes in response to sowing dates.**

| Treatments | Number of spikes per plant | Spike length (cm) | 1,000 seed weight (g) | Seed weight per spike (g) | Harvest index (%) | HUE (kg ha$^{-1}$ °C day$^{-1}$) | Grain filling Rate (kg ha$^{-1}$ day$^{-1}$) | Seed yield (kg/ha) |
|---|---|---|---|---|---|---|---|---|
| **Year (Y)** | | | | | | | | |
| 2021–22 | 31.5[a†] | 17.06[b] | 1.145[a] | 0.461[b] | 23.86[a] | 0.38[a] | 17.8[a] | 579.2[a] |
| 2022–23 | 27.0[b] | 18.24[a] | 1.143[a] | 0.474[a] | 17.72[b] | 0.32[b] | 13.4[b] | 494.6[b] |
| *P* value | <0.001 | <0.001 | NS | 0.004 | <0.001 | <0.001 | <0.001 | <0.001 |
| **Date of sowing (DOS)** | | | | | | | | |
| S1–1st July | 35.4[c] | 21.08[a] | 1.166[a] | 0.635[ab] | 17.54[de] | 0.32[g] | 17.5[cd] | 698.5[c] |
| S2–15th July | 36.2[b] | 21.33[a] | 1.166[a] | 0.652[a] | 19.24[d] | 0.36[f] | 18.3[b–d] | 755.5[b] |
| S3–1st August | 40.6[a] | 21.71[a] | 1.164[ab] | 0.654[a] | 21.22[cd] | 0.43[e] | 20.0[bc] | 811.0[a] |
| S4–15th August | 40.4[a] | 21.29[a] | 1.162[ab] | 0.651[a] | 21.92[b–d] | 0.48[d] | 19.5[bc] | 810.7[a] |
| S5–1st September | 38.7[ab] | 20.97[a] | 1.158[ab] | 0.611[bc] | 25.58[a–c] | 0.51[bc] | 23.9[a] | 793.1[a] |
| S6–15th September | 35.5[c] | 20.10[ab] | 1.155[a–c] | 0.578[c] | 26.47[ab] | 0.53[a] | 21.0[ab] | 741.0[b] |
| S7–1st October | 33.8[c] | 18.91[bc] | 1.154[a–d] | 0.543[d] | 26.38[ab] | 0.52[ab] | 21.1[ab] | 682.1[c] |
| S8–15th October | 30.4[d] | 18.19[cd] | 1.146[a–e] | 0.512[d] | 30.04[a] | 0.51[c] | 19.8[bc] | 613.6[d] |
| S9–1st November | 24.8[e] | 17.25[cd] | 1.145[a–e] | 0.465[e] | 28.72[a] | 0.48[d] | 18.4[bc] | 564.4[e] |
| S10–15th November | 24.0[e] | 16.49[de] | 1.143[b–e] | 0.436[e] | 27.55[a] | 0.42[e] | 15.4[d] | 477.6[f] |
| S11–1st December | 20.6[f] | 14.87[ef] | 1.135[c–e] | 0.390[f] | 24.97[a–c] | 0.29[h] | 20.8[b] | 393.7[g] |
| S12–15th December | 19.5[f] | 14.69[f] | 1.133[de] | 0.354[g] | 19.80[d] | 0.22[i] | 9.8[e] | 341.0[h] |
| S13–1st January | 20.6[f] | 14.26[f] | 1.126[ef] | 0.236[h] | 13.08[e] | 0.11[j] | 5.6[f] | 218.7[i] |
| S14–15th January | 19.9[f] | 13.45[f] | 1.107[fg] | 0.153[i] | 6.55[f] | 0.04[k] | 2.8[fg] | 105.3[j] |
| S15–1st February | 18.7[f] | 10.25[f] | 1.097[g] | 0.145[i] | 2.82[f] | 0.02[l] | 1.1[g] | 47.6[k] |
| *P* value | <0.001 | <0.001 | <0.001 | <0.001 | <0.001 | <0.001 | <0.001 | <0.001 |
| **Chia type (S)** | | | | | | | | |
| White | 28.3[b] | 17.31[b] | 1.137[b] | 0.418[b] | 21.32[a] | 0.33[b] | 14.8[b] | 509.2[b] |
| Black | 30.3[a] | 17.99[a] | 1.150[a] | 0.518[a] | 20.26[b] | 0.37[a] | 16.4[a] | 564.6[a] |
| *P* value | <0.001 | 0.003 | <0.001 | <0.001 | 0.0532 | <0.001 | 0.003 | <0.001 |
| **Interaction effect** | | | | | | | | |
| Y × S | 0.01 | NS | NS | NS | NS | <0.001 | NS | 0.004 |
| Y × DOS | <0.001 | <0.001 | NS | <0.001 | <.0001 | <0.001 | <0.001 | <0.001 |
| S × DOS | NS | NS | 0.001 | <0.001 | NS | <0.001 | NS | <0.001 |
| Y× S × DOS | NS | NS | NS | <0.001 | NS | <0.001 | NS | <0.001 |

**Notes.**
[†]Means followed by the same letter (s) within the column are not significantly differed ($p < 0.05$).
The different letter denoted in superscript in each column shows the significant difference between the treatments.

2021–22 black seeded chia produced higher seed yield in S3–S6, whereas in 2022–23 it was higher in sowings S3–S4 (Table S3). Therefore, sowing up to 15th September could favour the seed yield and heat use efficiency of chia morphotypes in semi-arid conditions.

### Weather parameters *vs* yield attributes of chia

Weather parameters across the growing period up to maturity established a significant ($p < 0.05$) relation with yield attributes of chia. The seed yield was positively influenced by RH ($r = 0.93$ and $R^2 = 0.856$), HUE ($r = 0.9$), and rainfall (RF) ($r = 0.76$ and $R^2 = 0.529$).

However, $T_{max}$ ($r = -0.82$ and $R^2 = 0.674$), $T_{diff}$ ($r = -0.87$ and $R^2 = 0.856$), accumulated GDD ($r = -0.31$), BSS ($r = -0.84$) were negatively influenced the seed yield (Figs. 4 and 5A–5D). Besides, $T_{max}$ is negatively related to chia yield attributes; seed yield per spike ($r = -0.76$), spike length ($r = -0.65$), and 1,000 seed weight ($r = -0.58$) (Fig. 4). Notably, RH during the entire cropping period displayed strong positive associations with chia yielding traits; ($r = 0.71$ to $0.85$). Analysis of diurnal temperature difference revealed a negative correlation with all growth and yield-related traits of chia. Moreover, the relation between seed yield and plant traits was also found to be significant (Fig. 4). Seed yield exhibited positive correlations with plant height ($r = 0.73$), spike length ($r = 0.78$), number of spikes per plant ($r = 0.80$), number of branches per plant ($r = 0.76$), seed yield per spike ($r = 0.89$), and 1,000 seed weight ($r = 0.74$). Conversely, seed yield showed a negative correlation with flowering duration ($r = -0.66$) and crop duration ($r = -0.28$). Hence, prevailing weather parameters had a considerable role in determining the growth and yield of chia morphotypes.

## Interaction between yield traits and weather parameters due to sowing dates and chia morphotypes

Multivariate analysis was conducted to elucidate the interaction, inter-relationship and variations among various yield traits and weather parameters that prevailed during the entire chia duration and chai types. Principal component analysis (PCA) revealed that the first two components (PC1 and PC2) captured 94.1% of the total variability (Fig. 6A). In PC1, traits such as seed yield, spike length, number of spikes, seed weight per spike, plant height, and biomass production demonstrated strong positive associations as indicated by the narrow angles between their vectors. Similarly, weather parameters RH, RF, and $T_{min}$ showed strong positive associations with seed yield and yield-related traits. These variables explained the maximum total variability as evidenced by the length of their vectors. Conversely, BSS, flowering duration, and $T_{max}$ exhibited negative associations with each other. In PC2, variables such as accumulated GDD, day length, flowering duration, days to 50% flowering, days to maturity, and FBA were positively associated with each other but negatively influenced the seed yield (Fig. 6A). Furthermore, sowing times (S4, S5, and S6) in both black and white seed types were closely related to higher seed yield, and yield traits as favoured by weather parameters like RH and rainfall. Conversely, delayed sowing times (S13, S14, and S15) in both varieties coincided with intense sunshine hours, poor RH, and higher $T_{max}$ resulting in longer flowering duration, and accumulated GDD negatively determined the seed yield and yield traits of chia (Fig. 6B).

## DISCUSSION

The deviation in ideal weather conditions due to changing sowing dates notably influences the flowering phenology, maturity and determines the yield of short day crops like chia. Therefore, this is a kind of first report that exhaustively analysed various dates of sowing and two chia morphotypes, establishing the relationship between weather and yield parameters.

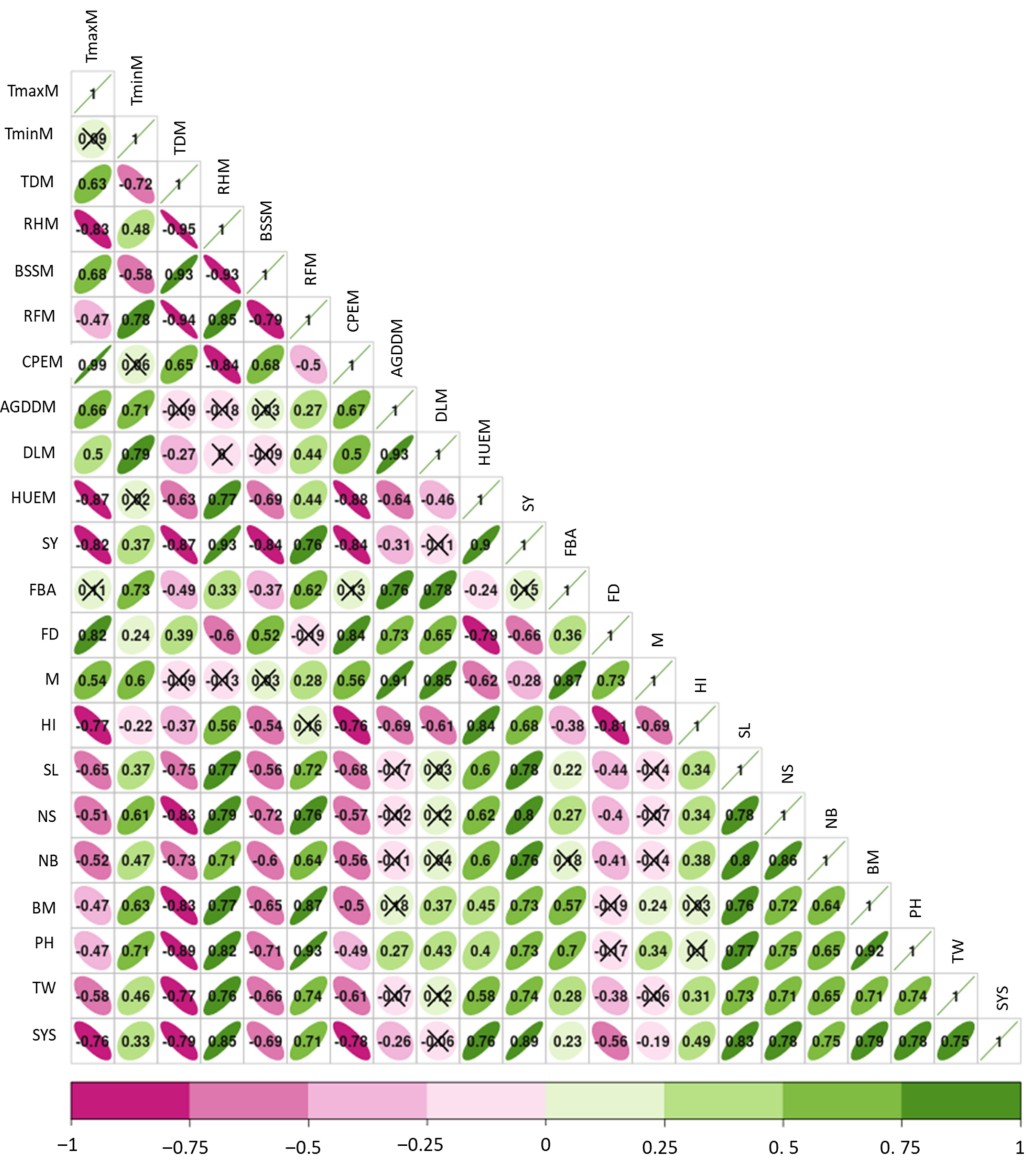

**Figure 4  Relation between seed yield, yield traits, weather parameters and flowering phenology of chia.** SY, seed yield; NS; number of spikes, TW; 1,000 seed weight, SL; spike length, BM; biomass accumulation, PH; plant height, TminM; minimum temperature, TmaxM; maximum temperature, TDM; temperature difference, RHM; relative humidity, BSSM; bright sun shine hours, RFM; rainfall, FBA; days to flower bud appearance; FD; flowering duration, M; days to maturity, AGDDM; accumulated growing degree days, DLM; day length, SYS; Seed yield per spike, CPEM; cumulative pan evaporation till maturity, HUE; heat use efficiency, HI; harvest index.

## Growth parameters of chia

Black seeded chia morphotype was found to be more vigorous over white seeded owing to greater plant height and biomass accumulation. This might be due to its superior genetic characteristics and adaptation as described by *Grimes et al. (2018)* and *Guttedar et al. (2023)*. Early sowing during the rainy season (July S1–S2) resulted in higher plant height

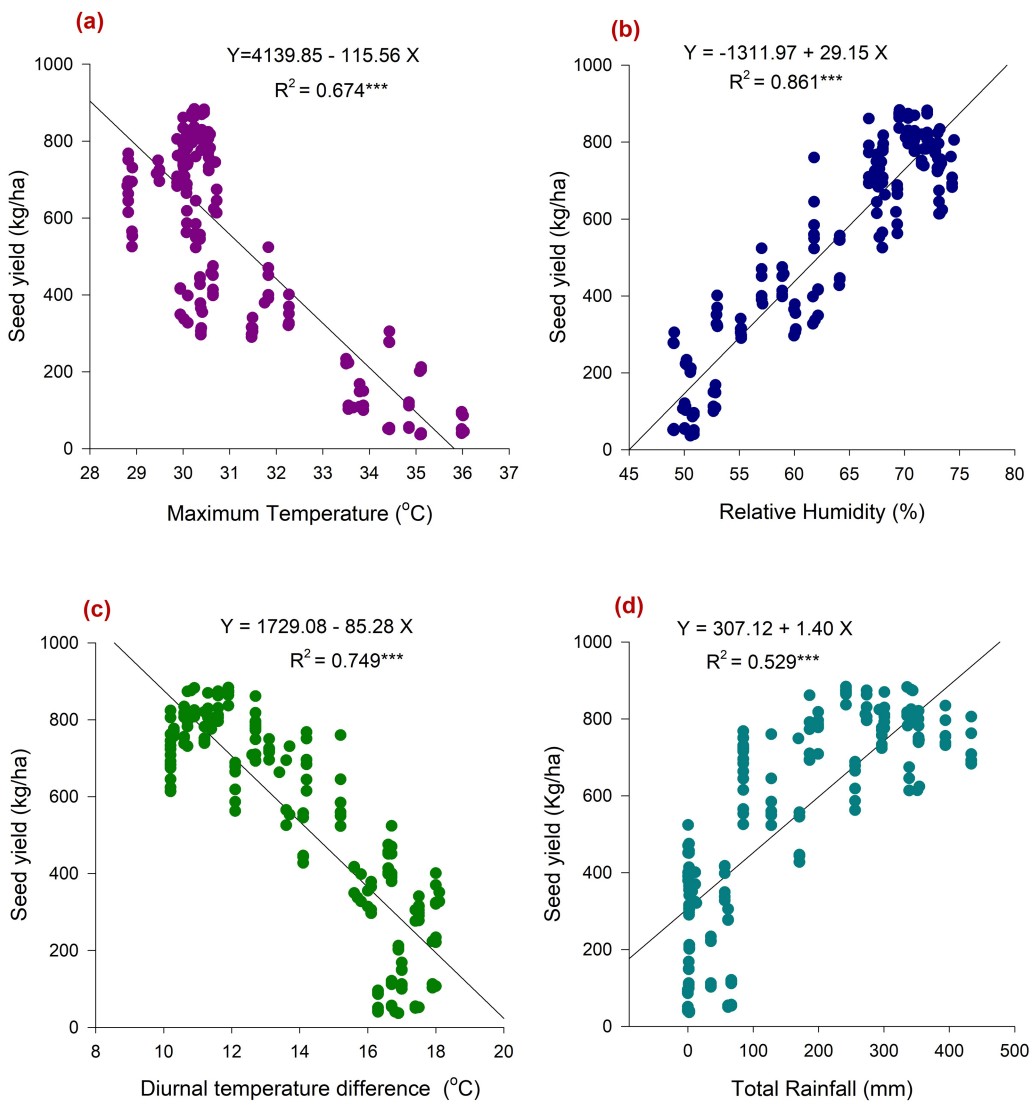

**Figure 5** **Association between weather parameters prevailed during the cropping period and seed yield of chia.** (A) maximum temperature during cropping period *vs* seed yield; (B) Relative humidity during cropping period *vs* seed yield; (C) diurnal temperature difference during cropping period *vs* seed yield; (D) total rainfall during cropping period *vs* seed yield.

and biomass accumulation because of long day conditions (average day length; 12.5 h and accumulated GDD; >2,000 °C) (Fig. 1C), led to more vegetative growth and delayed reproductive growth over subsequent sowing dates (*Guttedar et al., 2023*). This was also due to the receipt of sufficient rain, and the prevailing ideal temperature around ~30 °C during the vegetative phase favoured the growth and biomass accumulation in both types of chia, thus increasing the risk of lodging. Our findings corroborate the results of *Goergen et al. (2018)* in chia, where higher GDD and longer photoperiod increase plant height and biomass. Similarly, *Silva et al. (2018)* reported enhanced vegetative growth in chia due to a greater number of branches during early sowing. Whereas shorter plants with reduced

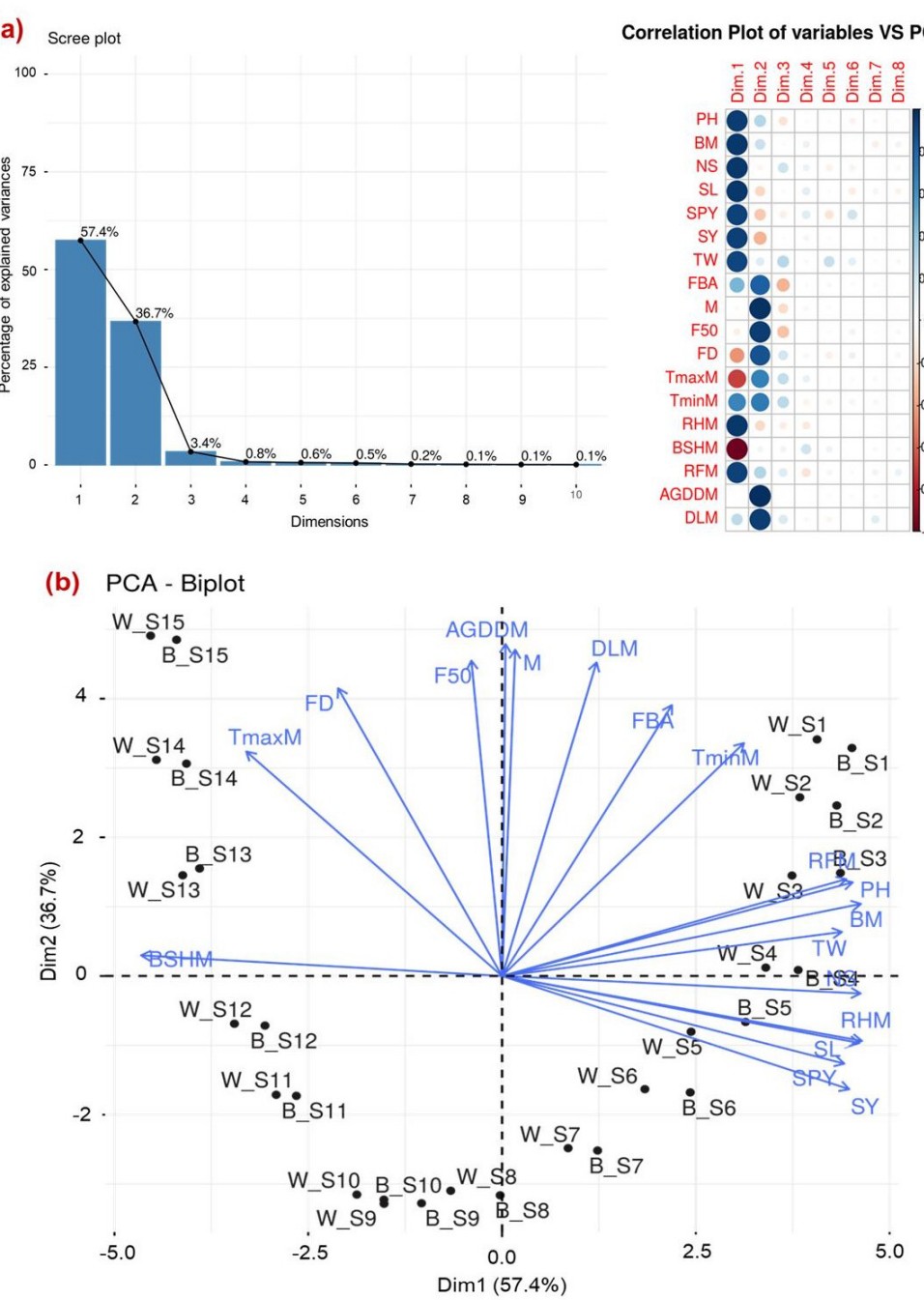

**Figure 6 Interrelation between seed yield and yield attributing traits across various sowing dates (A) PCA explaining the variance observed and contributing factors; (B) PCA-Biplot showing the interaction between traits across the variety and sowing dates.** W, white seed type; B, black seed type; S1–15, various sowing dates; SY, seed yield; SPY, seed per spike; NS, number of spikes; TW, test weight; SL, spike length, BM; biomass accumulation, PH; plant height, TminM; Minimum temperature, TmaxM; Maximum temperature, RHM; relative humidity, BSHM; bright sun shine hours, RFM; rainfall, FBA; days to flower bud appearance; F50; days to 50% flowering, FD; flowering duration, M; days to maturity, AGDDM; accumulated growing degree days, DLM; day length.

biomass accumulation in case of delayed sowings after December (S11) to February (S15) were attributed to prevailing dry weather (high temperature and low RH) with the least rainfall during active growth stages (Figs. 1A–1C). The crop biomass production is closely associated with dominant environmental factors such as temperature, RH, and rainfall, which together determine crop duration (*Guttedar et al., 2023*). Thus, chia is very sensitive to day length, RH, and temperature, which determines its biomass accumulation and yield.

## Flowering phenology and maturity in chia

The delayed FBA in chia during early (S1–S3) and delayed sowings (S14–S15) was possibly due to longer day length conditions (>12.5 h) compared to intermediate sowings (S4–S13) with shorter day lengths (<12 h). As a result, flowering duration was extended (56.6 to 77.5 days) owing to more number of days between FBA and completion of flowering. The positive correlation between the flowering duration and $T_{max}$ during flowering phase indicates the potential cause for delayed flower opening due to high temperatures (Fig. 4). It is important to note that hot weather (high temperature; >34 °C, low RH; <50%, and no rainfall; Fig. 1), along with the commencement of long days during flowering phase (March–April) in delayed sowing resulted in delayed FBA, and also the conversion of floral structures into vegetative parts in chia (*Guttedar et al., 2023*). Whereas, delayed flowering in early sowing (S1–S2) was probably related to long day conditions associated with higher RH, rainfall and accumulated GDD during the vegetative stage. Similarly, *Grimes et al. (2018)* and *Benetoli da Silva et al. (2020)* highlighted that alterations in chia phenology are primarily linked to fluctuations in RH and higher GDD. *Brandan, Curti & Acreche (2020)* also reported that the growing degree days are longer than the pre-flowering phase. Therefore, aligning chia flowering with optimal RH and rainfall conditions, synchronizing flower opening and ensuring a shorter flowering duration are crucial for efficient resource utilization and mitigating high temperatures and long days with higher accumulated GDD (*Foulkes et al., 2011*; *Sylvester-Bradley, Riffkin & O'Leary, 2012*).

Days taken for flower opening and its completion decide the duration of crop maturity. In the present study, early sowings as well as delayed sowings extended the chia maturity (125 to 143 days) compared to intermediate sowings (93 to 114 days), primarily due to delayed FBA, and flowering duration in chia. Similarly, *Lobo et al. (2011)* in Tucumán, Argentina and *Baginsky et al. (2016)* in Las Cruces, Chile, demonstrated that January sowing resulted in delayed flowering (105–111 days) and crop maturity (160–170 days respectively). Subsequently, grain filling duration (between 50% flowering and maturity) was extended with early and delayed sowing dates. A similar trend was observed with the completion of flowering (*Jamboonsri et al., 2012*; *Sattar et al., 2023*). Further, chia maturity was slightly delayed in 2022–23. This delay was likely attributed to increased accumulated GDD, higher RH and the occurrence of rainfall, which fostered enhanced vegetative growth. Thus the crop has taken more days to complete flowering and extended grain filling duration, as a positive correlation was observed between FBA, RH, rainfall, and day length (Fig. 3A). A similar pattern of extended maturity and grain filling duration was found in lentil (*Jamboonsri et al., 2012*; *Maphosa, Preston & Richards, 2023*). Both white and black seed chia types did not differ with respect to flowering phenology and maturity.

Therefore, chia, being a short-day tropical plant, thrives well under photoperiods of less than 12.5 h of light.

## Seed yield and yield attributes of chia

Black seeded chia morphotypes produced greater seed yield (564.6 kg ha$^{-1}$, 10.8% higher) than the white type (509.2 kg ha$^{-1}$). This improvement in seed yield with black types could be attributed to improved biomass accumulation and yield-contributing parameters such as number of spikes, spike length, and 1,000 seed weight (1.05 g). Previous researchers have noticed the genetic variation and superiority of black seeded chia types for yielding characters because of their wider adaptability (*Ayerza & Coates, 2009*; *Guttedar et al., 2023*). In this investigation, the seed yield of chia varied from 47.6 to 811.0 kg ha$^{-1}$ across fifteen sowing dates. The higher seed yield with mid sowing dates (S3–S5) was mainly due to improved yield contributing parameters (Table 2 and Fig. 3B). Similar associations between seed yield and traits such as the number of spikes, spike length, and harvest index have been reported in both black and white types of chia (*Baginsky et al., 2016*). The positive relation between flower dry weight and seed yield in chia was also reported by *Brandan, Curti & Acreche (2020)*. Despite congenial RH, rainfall and temperature, early sowing dates (S1 and S2) produced lower seed yield because the plants produced a lower number of spikes because of more height and canopy spread due to prolonged vegetative phase under long days. This might hinder the production of branches, inflorescences, and subsequent translocation of photosynthates towards seed filling. This concurs with the finding of *Han et al. (2006)* in soybean, where overcrowding canopy leads to poor branching with less number of pods. Interestingly, delayed sowing after S5 to S15 drastically reduced the seed yield of chia, ranging between 8.63–94.13% (Table 2). The poor chia seed yield was primarily due to the underdevelopment of yield governing traits, as reported by *Guttedar et al. (2023)*, that delayed sowing (October) reduced the seed yield in Indian conditions. Therefore, it is crucial to complete sowing by 1st August (S3) to 1st September (S5) to achieve higher seed yield (790–811 kg ha$^{-1}$) in chia.

## Weather and yield attributes of chia

This study among a few, clearly deciphered the impact of weather parameters in determining the chia yield across various (fifteen) sowing windows in a year. The biplot analysis confirmed that prevalence of optimum temperature (30–31 °C), rainfall (200–350 mm), and RH (67–72%) during S3–S5 sowing (1st August to 1st September) resulted in higher seed yield attributes (Fig. 6B). Our findings are in conformity with the results of *Grimes et al. (2018)* and *Benetoli da Silva et al. (2020)* that climatic requirements of moderate to high temperature, minimum temperature (<10 °C) with adequate rainfall enhanced the chia yield in Germany and Brazil. Meanwhile, delayed sowings (S10–S15) reduced the 1,000 seed weight of chia, mainly due to higher temperatures during the flowering phase, leading to prolonged flowering and grain-filling durations. This might also affect pollination, resulting in grain shrinkage due to the production of reactive oxygen species, reduced pollen tube development, increased pollen mortality, and grain abortion (*Nawaz et al., 2013*; *Dubey et al., 2019*). Thus, prolonged flowering and maturity durations negatively

influenced the seed yield, owing to non-synchronized flowering, resulting in poor seed setting and seed yield per spike, as evidenced by a negative correlation between seed yield per spike and flowering duration ($r = -0.56$). Therefore, the increased temperature during the grain filling period increases the percentage of chaffy seed formation, as found in cases of soybean and rice (*Borowska & Prusiński, 2021*; *Sanwong et al., 2023*).

Similarly, the reduced grain filling rate in chia might be due to increased thermal load, which is manifested in terms of higher accumulation of GDD leading to higher grain filling duration in early (S1–S4) and delayed sowings (S8–S15). These sowing dates also decreased the HUE in chia, despite higher seed yield in S3 and S4, poor HUE was possibly related to more accumulation of GDD. Similar findings were reported in wheat, where very early and delayed sowings significantly reduced the thermal use efficiency (*Kaur & Pannu, 2008*; *Singh, Kingra & Singh, 2016*). It's also found that exposing crops to higher temperatures at critical growth phases tends to affect the phenophase duration, HUE and yield (*Parya et al., 2010*). The sowing dates; S4–S6 result in a medium flowering phase (50–51 days), and maturity (98.8–114.4 days), leading to ideal plant height and biomass accumulation in chia over early and delayed sowing dates. Therefore, chia sowing from 1st August to 15th September coincides with optimum RH, lower diurnal temperature difference, and rainfall, favoured the yield attributes and seed yield of chia in semi-arid conditions.

## CONCLUSIONS

The study revealed that chia being a short-day plant, showed a significant response to different sowing windows. Weather conditions during the cropping period played a crucial role in chia's floral phenology, maturity, yield-contributing traits, and overall seed yield. Among chia morphotypes, black seed varieties exhibited greater vigour compared to white types. August 1st and September 1st was found to be optimal sowing times for both the chia types due to favourable weather conditions, including relative humidity (∼67–72%), maximum temperature (∼30–31 °C), day length (<12.0 h), rainfall (∼200–350 mm), and accumulated growing degree days (∼1,521–1,891). This sowing window of 30–45 days between 1st August to mid September in semi-arid regions can assist farmers in aligning chia cultivation with the desirable weather, cropping systems, and resource availability, thereby reducing climate-related risks. Extra-early sowing (July) reduces chia seed yield by 10.35%, moderately delayed sowing (September 15th to November 15th) by 24.1%, and extra-delayed sowing (December 1st to February 1st) resulting in a drastic reduction of 72.7%. Understanding these weather associations can support intensified chia cultivation practices. The findings suggest practical guidance for selecting suitable regions and optimal sowing dates for chia cultivation under evolving climate conditions, thereby contributing to sustainable development goals, particularly SDG 13 (climate action). Enhanced production also presents export opportunities to meet the growing industrial demand for chia seeds.

### Funding

This research was carried out in support of the ICAR-National Institute of Abiotic Stress Management in-house flagship project on "Augmenting farm income in water-scarce regions with alternative crops (IXX15656). The funders had no role in study design, data collection and analysis, decision to publish, or preparation of the manuscript.

### Grant Disclosures

The following grant information was disclosed by the authors:
ICAR-National Institute of Abiotic Stress Management: IXX15656.

### Competing Interests

The authors declare there are no competing interests. GR Halagundegowda is an employee of the Central Silk Board, India.

### Author Contributions

- CB Harisha conceived and designed the experiments, performed the experiments, analyzed the data, prepared figures and/or tables, authored or reviewed drafts of the article, and approved the final draft.
- KM Boraiah conceived and designed the experiments, performed the experiments, prepared figures and/or tables, authored or reviewed drafts of the article, and approved the final draft.
- PS Basavaraj conceived and designed the experiments, performed the experiments, authored or reviewed drafts of the article, and approved the final draft.
- Hanamant M. Halli conceived and designed the experiments, performed the experiments, analyzed the data, prepared figures and/or tables, and approved the final draft.
- Ram Narayan Singh analyzed the data, authored or reviewed drafts of the article, and approved the final draft.
- Jagadish Rane conceived and designed the experiments, analyzed the data, prepared figures and/or tables, and approved the final draft.
- K Sammi Reddy conceived and designed the experiments, prepared figures and/or tables, and approved the final draft.
- GR Halagundegowda analyzed the data, authored or reviewed drafts of the article, and approved the final draft.
- Amresh Chaudhary conceived and designed the experiments, analyzed the data, prepared figures and/or tables, authored or reviewed drafts of the article, and approved the final draft.
- Arvind Kumar Verma analyzed the data, authored or reviewed drafts of the article, and approved the final draft.
- Y Ravi analyzed the data, authored or reviewed drafts of the article, and approved the final draft.

- Honnappa Asangi analyzed the data, authored or reviewed drafts of the article, and approved the final draft.
- E Senthamil performed the experiments, analyzed the data, prepared figures and/or tables, authored or reviewed drafts of the article, and approved the final draft.

## Data Availability

The raw data is available in the Supplementary File.

## Supplemental Information

Supplemental information for this article can be found online at http://dx.doi.org/10.7717/peerj.19210#supplemental-information.

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

# PeerJ

**Brandan JP, Curti RN, Acrechea MM. 2019.** Phenological growth stages in chia (*Salvia hispanica* L.) according to the BBCH scale. *Scientia Horticulturae* **255**:292–297 DOI 10.1016/j.scienta.2019.05.043.

**Brandan JP, Izquierdo N, Acreche MM. 2022.** Oil and protein concentration and fatty acid composition of chia (*Salvia hispanica* L.) as affected by environmental conditions. *Industrial Crops and Products* **177**:114496 DOI 10.1016/j.indcrop.2021.114496.

**Capitani MI, Ixtaina VY, Nolasco SM, Tomás MC. 2013.** Microstructure, chemical composition and mucilage exudation of chia (*Salvia hispanica* L.) nutlets from Argentina. *Journal of the Science of Food and Agriculture* **93(15)**:3856–3862 DOI 10.1002/jsfa.6327.

**ChiaSeedMarket. 2024.** Future market insights. *Available at* https://www.futuremarketinsights.com/reports/chia-seed-market (accessed on 04 April 2024).

**Dubey R, Pathak H, Singh S, Chakraborty B, Thakur AK, Fagodia RK. 2019.** Impact of sowing dates on terminal heat tolerance of different wheat (*Triticum aestivum* L.) cultivars. *National Academy Science Letters* **42**:445–449 DOI 10.1007/s40009-019-0786-7.

**Felisberto MHF, Galvão MTEL, Picone CSF, Cunha RL, Pollonio MAR. 2015.** Effect of prebiotic ingredients on the rheological properties and microstructure of reduced-sodium and low-fat meat emulsions. *LWT - Food Science and Technology* **60**:148–155 DOI 10.1016/j.lwt.2014.08.004.

**Fernandes SS, Romani VP, Filipini GS, Martins VG. 2020.** Chia seeds to develop new biodegradable polymers for food packaging: properties and biodegradability. *Polymer Engineering and Science* **60(9)**:1–10 DOI 10.1002/pen.25464.

**Foulkes MJ, Slafer GA, Davies WJ, Berry PM, Sylvester-Bradley R, Martre P, Calderini DF, Griffiths S, Reynolds MP. 2011.** Raising yield potential of wheat, III. Optimizing partitioning to grain while maintaining lodging resistance. *Journal of Experimental Botany* **62(2)**:469–486 DOI 10.1093/jxb/erq300.

**Goergen PCH, Nunes UR, Stefanello R, Lago I, Nunes AR, Durigon A. 2018.** Yield and physical and physiological quality of *Salvia hispanica* L. seeds grown at different sowing dates. *Journal of Agricultural Science* **10(8)**:182–191 DOI 10.5539/jas.v10n8p182.

**Gomez KA, Gomez AA. 1984.** *Statistical procedures for agricultural research*. 2nd edition. New York: John Wiley and Sons, 680.

**Gopinath PP, Prasad R, Joseph B, Adarsh VS. 2021.** grapesAgri1: collection of shiny apps for data analysis in agriculture. *Journal of Open Source Software* **6(63)**:3437 DOI 10.21105/joss.03437.

**Grimes SJ, Phillips TD, Hahn V, Capezzone F, Graeff-Hönninger S. 2018.** Growth, yield performance and quality parameters of three early flowering chia (Salvia hispanica L.) genotypes cultivated in Southwestern Germany. *Agriculture* **8(10)**:154 DOI 10.3390/agriculture8100154.

**Guttedar A, Hunje R, Badiger B, Sharma Y. 2023.** Effect of dates of sowing on seed yield and quality of chia (*Salvia hispanica* L.). *The Pharma Innovation Journal* **12(11)**:1857–1861.

**Halli HM, Govindasamy P, Wasnik VK, Shivakumar BG, Swami S, Choudhary M, Yadav VK, Singh AK, Raghavendra N, Govindasamy V, Chandra A. 2024.** Climate-smart deficit irrigation and nutrient management strategies to conserve energy, greenhouse gas emissions, and the profitability of fodder maize seed production. *Journal of Cleaner Production* **442**:140950 DOI 10.1016/j.jclepro.2024.140950.

**Han T, Cunxiang W, Tong Z, Mentreddy R, Tan T, Gai J. 2006.** Postflowering photoperiod regulates vegetative growth and reproductive development of soybean. *Environmental and Experimental Botany* **55**:120–129 DOI 10.1016/j.envexpbot.2004.10.006.

**Harisha CB, Narayanpur VB, Rane J, Ganiger VM, Prasanna SM, Vishwanath YC, Reddi SG, Halli HM, Boraiah KM, Basavaraj PS, Eman A, Mahmoud EA, Casini R, Elansary HO. 2023.** Promising bioregulators for higher water productivity and oil quality of chia under deficit irrigation in semiarid regions. *Plants* **12(3)**:662 DOI 10.3390/plants12030662.

**Harisha CB, Rane J, Halagunde Gowda GR, Chavan SB, Chaudhary A, Verma AK, Ravi Y, Asangi H, Halli HM, Boraiah KM, Basavaraj PS, Kumar P, Reddy KS. 2024.** Effect of deficit irrigation and intercrop competition on productivity, water use efficiency and oil quality of chia in semi-arid regions. *Horticulturae* **10**:101 DOI 10.3390/horticulturae10010101.

**Hirich A, Choukr-Allah R, Jacobsen SE. 2014.** Quinoa in Morocco-effect of sowing dates on development and yield. *Journal of Agronomy and Crop Science* **200**:371–377 DOI 10.1111/jac.12071.

**Jamboonsri W, Phillips T, Geneve R, Cahill J, Hildebrand D. 2012.** Extending the range of an ancient crop, Salvia hispanica L. - a new $\omega$3 source. *Genetic Resources and Crop Evolution* **59**:171–178 DOI 10.1007/s10722-011-9673-x.

**Jingar A, Preet MS, Kumar A, Ram M. 2023.** Response of date of sowing and crop geometry on yield potential of chia under tropical conditions. *Biological Forum –An International Journal* **15(8)**:470–473.

**Karim MM, Ashrafuzzaman M, Hossain MA. 2015.** Effect of planting time on the growth and yield of chia (*Salvia hispanica* L.). *Asian Journal of Medical and Biological Research* **1(3)**:502–507 DOI 10.3329/ajmbr.v1i3.26469.

**Kaur A, Pannu RK. 2008.** Effect of sowing time and nitrogen schedules on phenology, yield and thermal-use efficiency of wheat (*Triticum aestivum* L.). *Indian Journal of Agricultural Sciences* **78(4)**:366–369.

**Kris-Etherton PM, Krauss RM. 2020.** Public health guidelines should recommend reducing saturated fat consumption as much as possible: YES. *The American Journal of Clinical Nutrition* **112(1)**:13–18 DOI 10.1093/ajcn/nqaa110.

**Lobo R, Alcocer M, Fuentes FJ, Rodríguez WA, Morandini M, Devani MR. 2011.** Chia crop production in Tucuman, Argentina. *EEAOC—Avance Agro industrial* **32(4)**:27–30.

**Maphosa L, Preston A, Richards MF. 2023.** Effect of sowing date and environment on phenology, growth and yield of lentil (*Lens culinaris* Medikus) Genotypes. *Plants* **12**:474 DOI 10.3390/plants12030474.

Nawaz A, Farooq M, Cheema SA, Wahid A. 2013. Differential response of wheat cultivars to terminal heat stress. *International Journal of Agriculture and Biology* **15(6)**:1354–1358.

Nuttonson MY. 1957. Wheat-climate relationship and the use of phenology in ascertaining the thermal and photo thermal requirement of wheat. *BioScience* **7(1)**:39–40 DOI 10.2307/1292056.

Parya M, Nath D, Mazumdar D, Chakraborty PK. 2010. Effect of thermal stress on wheat productivity in West Bengal. *Journal of Agrometeorology* **12(2)**:217–220 DOI 10.54386/jam.v12i2.1309.

Pathak PO, Nagarsenker MS, Barhate CR, Padhye SG, Dhawan VV, Bhattacharyya D, Viswanathan CL, Steiniger F, Fahr A. 2015. Cholesterol anchored arabinogalactan for asialoglycoprotein receptor targeting: synthesis, characterization, and proof of concept of hepatospecific delivery. *Carbohydrate Research* **408**:33–43 DOI 10.1016/j.carres.2015.03.003.

Rajagopal V, Choudhary RL, Kumar N, Krishnani KK, Singh Y, Bal SK, Minhas PS, Singh NP. 2018. Soil health status of NIASM southern farm land. In: *ICAR–National Institute of Abiotic Stress Management, Baramati, India*, 51–52.

Ram M, Meena RC, Ambawat S, Bhardwaj R, Kumar M, Meena DS, Kumawat R, Choudhary S. 2024. Chia (*Salvia hispanica* L.) production potential in western India influenced by planting date and crop geometry. *International Journal of Research in Agronomy* **7(1)**:277–282 DOI 10.33545/2618060X.2024.v7.i1d.228.

Ray DK, West PC, Clark M, Gerber JS, Prishchepov AV, Chatterjee S. 2019. Climate change has likely already affected global food production. *PLOS ONE* **14(5)**:e0217148 DOI 10.1371/journal.pone.0217148.

Saini RK, Keum YS. 2018. Omega-3 and omega-6 polyunsaturated fatty acids: dietary sources, metabolism, and significance—a review. *Life Sciences* **203**:255–267 DOI 10.1016/j.lfs.2018.04.049.

Sanwong P, Sanitchon J, Dongsansuk A, Jothityangkoon D. 2023. High temperature alters phenology, seed development and yield in three rice varieties. *Plants* **12**:666 DOI 10.3390/plants12030666.

Sattar A, Nanda G, Singh G, Jha RK, Bal SK. 2023. Responses of phenology, yield attributes, and yield of wheat varieties under different sowing times in Indo-Gangetic Plains. *Frontiers in Plant Science* **14**:1224334 DOI 10.3389/fpls.2023.1224334.

Silva BP, Anunciaçao PC, Matyelka JCDS, Della Lucia CM, Martino HSD, Pinheiro-Sant'Ana HM. 2017. Chemical composition of Brazilian chia seeds grown in different places. *Food Chemistry* **221**:1709–1716 DOI 10.1016/j.foodchem.2016.10.115.

Silva H, Arriagada C, Campos-Saez S, Baginsky C, Castellaro-Galdames G, Morales-Salinas L. 2018. Effect of sowing date and water availability on growth of plants of chia (*Salvia hispanica* L.) established in Chile. *PLOS ONE* **13(9)**:e0203116 DOI 10.1371/journal.pone.0203116.

Singh M, Khushu MK. 2012. Growth and yield prediction of wheat in relation to agroclimatic indices under irrigated and rainfed condition. *Journal of Agrometeorology* **14(1)**:63–66 DOI 10.54386/jam.v14i1.1386.

**Singh S, Kingra PK, Singh SP. 2016.** Heat unit requirement and its utilization efficiency in wheat under different hydrothermal environments. *Annals of Agricultural Research* **37(2)**:130–140.

**Suri S, Passi SJ, Goyat J. 2016.** Chia Seed (*Salvia hispanica* L.). A new age functional food. *International Journal of Advanced technology in Engineering and Science* **4(3)**:286–299.

**Sylvester-Bradley R, Riffkin P, O'Leary G. 2012.** Designing resource efficient ideotypes for new cropping conditions: wheat (*Triticum aestivum* L.) in the High Rainfall Zone of southern Australia. *Field Crops Research* **125**:69–82 DOI 10.1016/j.fcr.2011.07.015.