# Peer review of "Optimizing sowing time and weather conditions for enhanced growth and seed yield of chia (Salvia hispanica L.) in semi-arid regions"

_PeerJ, doi:10.7717/peerj.19210_

## Round 0.1 · original submission · Major Revisions

Dear authors, I kindly ask you to carefully improve the manuscript in accordance with the recommendations of anonymous reviewers. Experimental design methods are widely used in agriculture. Your study is an example of insufficient consideration of all the factors that influenced the studied crop. I hope that this article will generate significant interest among readers. I hope that eliminating the shortcomings indicated in the reviews of anonymous reviewers will help you successfully pass the second review and publish an improved version of this manuscript.

·

Basic reporting

Dear Authors,
Thanks for choosing such a wonderful topic to work on it. You have written the manuscript very well. I did not find much more inconsistency with the scientific backgrounds and the justification. However, I have some suggestions for further improvement. These are as below:

In line 47 delete the portion of the sentences of ....Understanding the .........conditions...... rather start with .................. The present study findings can help to......

Check the key word : "Growing degree day"
Please check the figure 4 and 5; if possible then merge the figure 5 with the figure 4.

Experimental design

No comment.

Validity of the findings

If possible please check the coherence of the results and successive discussion to avoid the overlap of the concept of the findings.

Additional comments

If possible, you can add some pictorial view of the experiment as supplementary materials.
The relationship between seed yield and weather data might be visualized in correlation matrix as well.
Try to focus on the precise and specific concluding remarks as per following the objectives of the study.


Thanks once again for your efforts to make a wonderful manuscript.

Reviewer 2 ·

Basic reporting

Basic reporting
Satisfactory.
The manuscript presents good information on "Optimizing sowing time and weather conditions for enhanced growth and seed yield of chia (Salvia hispanica L.) in semi-arid regions". The language is easy to understand and proper context is provided.
Introduction
Comment 1: The introduction needs more detail.
Comment 2: I suggest in the introduction add the area, production and productivity of chia in our country, this shows the relevance of research in relation to farming community.
Comment 3: 58-59: Provide a reference for the statement.
Comment 4: 101- Provide a reference for the statement.

Experimental design

Experimental design
No comment

Validity of the findings

Validity of the findings
No comment

Additional comments

Discussion
The results are aptly tabulated and expressed and the units of measurement are properly cited. The discussion is appropriate and supported by the research work done by the previous workers.
Comment 5: 319-321: Mention any related reference for the findings.
Comment 6: 326-328: The reason included in this lines and the reason from reference (334) are not same.
Tables and Figures
Comment 7: Table-2: Mention the reason for significantly higher seed yield in 2021-22 when compared to 2022-23.
Comment 8: Figure-1: Mention the title of fig 1a and 1b.
Comment 9: Figure-2: Mention the title of fig 2a and 2b.
Comment on language and grammar issues
Some examples where the language and grammar could be improved in the below mentioned lines
Comment 10: 47- Close the bracket for GDD.
Comment 11: 55- Check the sentence formation.
Comment 12: 70- Check the sentence formation.
Comment 13: 83- Check the sentence formation.
Comment 14: 85- Check the sentence formation.
Comment 15: 117- Check the sentence formation.
Comment 16: 136- Check the sentence formation.
Comment 17: 241- Check the sentence formation.
Comment 18: 246- Check the sentence formation.
Comment 19: 249- Check the sentence formation.
Comment 20: 292- Check the sentence formation.
Comment 21: 355- Check the sentence formation.

Annotated reviews are not available for download in order to protect the identity of reviewers who chose to remain anonymous.

Reviewer 3 ·

Basic reporting

This study explores the effects of environmental conditions on phenology, biomass partitioning, and yield in chia, presenting an experiment based on staggered sowing dates conducted over two consecutive years to assess the performance of two chia morphotypes. The authors link the duration of developmental phases and yield-related traits with the climatic conditions experienced by the crop on different sowing dates.

Experimental design

A) Analytical Issues
1. Experimental Design:
o The authors describe their approach as a factorial experiment; however, it is, in fact, a split-plot experiment. The term "sub-factor" should be corrected to "sub-plot."
o The authors assigned the chia morphotypes to the main plots and the sowing dates to the sub-plots. This approach should be reconsidered, as the sowing date does not function as a randomization factor. Instead, the chia morphotypes should be randomized within the sub-plots to adhere to sound experimental design principles.
2. Data Analysis:
o The analysis presented as a classical factorial experiment should instead reflect a combined analysis of variance for a split-plot design across years. Results for blocks nested within years, as well as main plot and sub-plot errors, are not included and should be presented.
3. Model Specification:
o The model used for the effects requires revision. The authors assume all effects, except for the block, are fixed. They should also consider the year as a random effect, as environmental conditions varied between years.
4. Interaction Effects:
o Results indicate significant year-by-sowing date interactions, yet the authors focus on main effects in isolation. Triple interactions are observed in some cases but are not analyzed in depth.
5. Variable Relationships:
o Several relationships presented do not appear to be linear correlations; some suggest curvilinear trends that should be explicitly addressed.
6. GGE Analysis:
o The multi-environment analysis using the GGE methodology is unclear. The authors should elaborate significantly, as GGE is typically used for interaction effects, while the authors are presenting inter-variable relationships.
B) Methodological Issues
1. Environmental Conditions:
o Given the environmental conditions reported, the authors should clarify how chia morphotypes were established in months without precipitation. It is essential to state explicitly that all experiments were conducted under rainfed conditions. Additionally, the distinction should be made between emergence and germination, as the former is being observed.
2. Humidity-Related Observations:
o The authors claim that relative humidity strongly influences both grain and biological yield in chia. However, they did not measure any indirect attributes to support this claim, such as stomatal conductance or water-use efficiency. This raises questions about the validity of their conclusions.
o There is no mention of phase durations correlated with environmental variables such as precipitation or maximum temperature. Furthermore, critical temperature thresholds for chia growth and development during vegetative and pre-flowering stages remain undetermined in this study.

Validity of the findings

In conclusion, while the manuscript provides valuable insights into chia cultivation under varying environmental conditions, significant revisions are necessary to enhance its scientific rigor and methodological clarity. I hope these comments will help the authors improve the quality of their work and align it with the journal’s publication standards.

Reviewer 4 ·

Basic reporting

.

Experimental design

.

Validity of the findings

.

Annotated reviews are not available for download in order to protect the identity of reviewers who chose to remain anonymous.

---

## Round 0.2 · Minor Revisions

Dear authors, I ask you to correct some minor comments and I believe that after this the manuscript can be recommended for publication.

·

Basic reporting

Authors revised correctly.

Experimental design

Satisfied with the experimental design and statistical analyses.

Validity of the findings

Tried to relate with the previous findings to validate their present findings.

Additional comments

Dear Authors, Thanks for your tremendous efforts to revise the manuscript with following all the comments and suggestions of the reviewers. However, I have found there are some minor corrections that need to be clarified further before final publication. I hope you will consider this correction further. Wishing you all the best of your manuscript.

Reviewer 2 ·

Basic reporting

Satisfactory and no comments

Experimental design

No comments

Validity of the findings

No comments

Additional comments

The title and the flow of presentation is clear, relevant, and scientifically significant, addressing the crucial need for optimized sowing strategies to enhance chia yield in semi-arid regions. The research has strong practical implications for farmers and agricultural planners, helping them make informed decisions for better yield and resource efficiency.

---

## Round 0.3 · Minor Revisions

Dear authors, I congratulate you on the acceptance of your article by the reviewer

Before final acceptance, a Section Editor noted:

> Overall this manuscript is well-written but I think English wording of the abstract can be substantially improved to increase readability and to reduce repetition. Because there were several grammatical errors as well, I used track changes on the Word doc to suggest changes throughout the abstract (attached). Authors may chose different wording from my suggestions; in that case authors should carefully check their wording in each sentence of the abstract when uploading their final version

·

Basic reporting

Thanks a lot to the authors for your efforts to revise the manuscript with taking all the comments in the review process. Now the manuscript found as worthy for publication in PeerJ.

Experimental design

No comment.

Validity of the findings

No comments.

Additional comments

No comments.

---

## Round 0.4 · accepted · Accept

Dear authors, I congratulate you on the acceptance of this manuscript for publication.